# Graphene-Based ESD Protection for Future ICs

**DOI:** 10.3390/nano13081426

**Published:** 2023-04-20

**Authors:** Cheng Li, Zijin Pan, Weiquan Hao, Xunyu Li, Runyu Miao, Albert Wang

**Affiliations:** Department of Electrical and Computer Engineering, University of California, Riverside, CA 92521, USA

**Keywords:** ESD protection, gNEMS, interconnects, switch, graphene, TLP, VFTLP, HBM, CDM

## Abstract

On-chip electrostatic discharge (ESD) protection is required for all integrated circuits (ICs). Conventional on-chip ESD protection relies on in-Si PN junction-based device structures for ESD. However, such in-Si PN-based ESD protection solutions pose significant challenges related to ESD protection design overhead, including parasitic capacitance, leakage current, and noises, as well as large chip area consumption and difficulty in IC layout floor planning. The design overhead effects of ESD protection devices are becoming unacceptable to modern ICs as IC technologies continuously advance, which is an emerging design-for-reliability challenge for advanced ICs. In this paper, we review the concept development of disruptive graphene-based on-chip ESD protection comprising a novel graphene nanoelectromechanical system (gNEMS) ESD switch and graphene ESD interconnects. This review discusses the simulation, design, and measurements of the gNEMS ESD protection structures and graphene ESD protection interconnects. The review aims to inspire non-traditional thinking for future on-chip ESD protection.

## 1. Introduction

ESD failure is a major reliability problem for ICs and microsystems. On-chip ESD protection is hence required for all ICs [1,2,3]. ESD failures occur due to a large ESD current transient that overheats an IC locally and/or a strong ESD voltage surge that causes the breakdown of an IC [1]. Figure 1 depicts a classic full-chip ESD protection scheme in which ESD protection devices are placed at each pad with respect to the ground (GND) and power supplies (e.g., V_DD_ and V_SS_) [1]. When an ESD transient appears at one pad, the normally OFF ESD protection structure will be swiftly turned ON to form a low-resistance (low-R) conduction path to discharge the large incident ESD pulse, thus protecting ICs. Theoretically, an ESD protection device behaves as a fast switch, which remains OFF during normal IC operations, not interfering with IC functions. The ESD switch can be quickly turned ON by an incoming ESD pulse to provide a low-R channel to discharge the large ESD pulse into local GND. Figure 2 shows a typical snapback I–V behavior for an ESD protection device, which is characterized by the critical parameters of ESD that include the ESD triggering threshold (voltage, V_t1_; current, I_t1_; and response time, t_1_); the ESD holding threshold (voltage, V_h_, and current, I_h_); ESD resistance (R_ON_); and the ESD thermal breakdown point (voltage, V_t2_, and current, I_t2_) [4]. A robust ESD protection device must be carefully designed for these critical parameters to comply with an ESD design window, as shown in Figure 2 [1]. As IC technologies advance with aggressive scaling, the device breakdown voltage (BV) dramatically decreases, while the supply voltage reduces only slightly, which leads to a reduction the ESD design window and thus makes advanced ESD protection design very challenging [1]. For decades, on-chip ESD protection has relied on PN-based devices that reside inside Si substrates (in-Si), such as diodes, bipolar junction transistors (BJTs), metal-oxide-semiconductor field-effect transistors (MOSFETs), silicon-controlled rectifiers (SCRs), and their derivatives or combinations (Figure 3) [1]. Such conventional in-Si PN-based ESD protection structures unavoidably introduce significant ESD protection design overhead problems, which include parasitic capacitance (C_ESD_), leakage current (I_leak_), and noises, as well as large area consumption and difficulty in chip layout floor planning [5,6]. The ESD-induced parasitic effects can severely affect IC performance. For example, a C_ESD_ of a few tens of pF will seriously affect almost all specs of radio-frequency (RF) ICs [7,8,9,10]. Such ESD-induced design overhead effects, inherent to in-Si PN-based ESD protection structures, are becoming increasingly unacceptable to advanced ICs in advanced technology nodes, e.g., millimeter-wave ICs for 5G wireless systems and high-data-rate serializer/deserializer (SerDes) ICs for communications. This, therefore, calls for truly disruptive ESD protection solutions for future ICs in nanonodes and beyond.

## 2. gNEMS ESD Switch

### 2.1. ESD Protection Structure and Mechanism

To overcome the fundamental challenge associated with conventional in-Si PN-based ESD protection structures, a novel above-IC graphene-based NEMS switch device structure was proposed and demonstrated experimentally [11]. The new gNEMS ESD switch structure is illustrated in Figure 4, which is a two-terminal device containing a suspended graphene nanoribbon over a cavity in a substrate. The two electrodes, i.e., the anode (A) and the cathode (K), are electrically separated by the cavity, hence in a normally OFF state. For on-chip ESD protection, a gNEMS device is connected to pads on an IC chip similar to that used in conventional in-Si PN-based ESD protection structures. An OFF gNEMS ESD switch will not affect normal IC operations. During an ESD event, when an ESD transient appears at one pad, the strong, transient electrical field generated will pull the suspended graphene film downward toward the bottom of the cavity. When the graphene film touches the cathode, it turns the gNEMS ON and forms a low-R conduction path to discharge the incident ESD pulse and hence provides ESD protection. After the ESD transient is over, the strong elastic force of the graphene film will pull itself upward and return to its suspension state, thus turning the gNEMS switch OFF [11,12]. Graphene materials are considered for ESD protection due to their material properties, such as extremely high carrier mobility (~15,000 cm^2^/Vs), very high thermal conductivity (4.84~5.30 × 10^3^ W/m), high Young’s modulus, superior mechanical strength, and superlight weight [13,14,15], all of which are desirable for ESD protection functions. For example, high carrier mobility results in low ESD resistance. Good thermal conductivity reduces ESD-induced overheating. High Young’s modulus and light weight ensure the fast switching speed of gNEMS devices. Strong mechanical strength increases the reliability of gNEMS ESD switches.

Compared with conventional in-Si PN-based ESD protection structures, this new gNEMS ESD switch device has several novelties and advantages [12]: First, the gNEMS device contains an air cavity and does not have any PN junction, which minimizes the ESD-induced parasitic capacitance and leakage current, ideally C_ESD_ ⇒ 0F and I_leak_ ⇒ 0A. This attribute is critically beneficial to advanced ICs because parasitic C_ESD_ can seriously affect RF ICs, and I_leak_ increases standby power consumption. Second, the ultra-high carrier mobility of graphene means the gNEMS device can carry more ESD current without overheating at a faster speed, translating into high ESD protection capability. Third, superior thermal conductivity facilitates the removal of the ESD-induced heat and hence enhances ESD robustness. Fourth, gNEMS is made in the back-end of the line (BEOL) in a complementary metal oxide semiconductor (CMOS), instead of residing inside a Si substrate, which theoretically removes the troublesome ESD-induced design overhead effects that are inherent to in-Si PN-based ESD protection structures. This novel above-IC gNEMS ESD switch structure can not only minimize the PN-induced ESD parasitic effects but also (ideally) consume no extra Si die area, which will also make IC layout floor planning much easier. Fifth, a gNEMS device is ideally a symmetric structure that can discharge ESD pulses in both directions, which can dramatically reduce the total head count of ESD protection devices on a chip required to form an ESD protection network. Sixth, a gNEMS device can be fabricated using CMOS-compatible processes. Overall, this disruptive gNEMS ESD protection device concept has the potential to revolutionize the ESD protection design field in the future.

### 2.2. gNEMS Fabrication

A fabrication procedure, depicted in Figure 5, was developed for gNEMS devices considering both CMOS process compatibility and 3D heterogeneous integration for future ICs. Figure 5a illustrates the five key processing steps for fabricating gNEMS devices. The substrate used is a heavily P-doped silicon wafer. First, low-pressure chemical vapor deposition (LPCVD) is used to deposit ~250 nm thick silicon dioxide (SiO_2_) above the doped silicon wafer as the main dielectric layer. Second, a thin layer (100 nm) of silicon nitride (Si_3_N_4_) is deposited using the plasma-enhanced chemical vapor deposition (PECVD) method as a hard mask for hydrogen fluoride (HF) etching. After Si_3_N_4_ deposition, reactive ion etching (RIE) is used to open a trench window in the substrate. Next, a graphene film is grown on the copper via chemical vapor deposition (CVD) with the process optimized for fabricating large-area graphene films for production. Figure 5b illustrates the Ramen spectra of both single-crystalline and poly-crystalline graphene films generated, in which the G and 2D peaks confirm the graphene structure, while the D peaks distinguish the structures of poly-crystalline and single-crystalline graphene materials. In the next step, the graphene film is transferred to the Si substrate, and oxygen plasma etching is used to pattern the graphene film into individual ribbons for gNEMS devices. Next, Ti/Pd/Au films (5/30/50 nm) are deposited using an e-beam, followed by a lift-off process to form the top electrodes. In the last step, the HF vapor method is used to etch off the SiO_2_ within the Si_3_N_4_ widow and release the graphene ribbons. Figure 5c shows a 3D scanning image for a fabricated gNEMS device structure, in which the suspended graphene ribbon can be readily observed [11,12].

### 2.3. Simulation Study of gNEMS Devices

To understand the gNEMS ESD protection mechanism and guide gNEMS design, a finite element method (FEM)-based simulation was conducted. Figure 6 shows a gNEMS ESD device in an ESD-test setting using a transmission line pulse (TLP) ESD stress tester. The simulated gNEMS had a length of (L) = 20 µm, a width of (W) = 10 µm, and a cavity depth of d = 350 nm. Figure 6a shows the vertical physical displacement (*Z*-axis) of the graphene ribbon at the moment when the suspended graphene ribbon touched the bottom electrode. The vertical displacement of the graphene ribbon is scaled in colors, with blue for “0” displacement (i.e., the suspended graphene film in its original position) and red for the largest bending displacement at the center (−350 nm bending). The bending and contact of the graphene ribbon appear to be uniform across the ribbon width. Figure 6b depicts the simulated vertical displacement characteristics of the graphene ribbon in the time domain during the TLP stressing period under a square pulse waveform of 7.2 V. It is readily observed that the suspended graphene membrane has the largest bending at the center, and the physical displacement increases as the TLP pulse continues in the time domain until touching the bottom. The simulation shows that as the time elapses, the electrostatic force induced by the TLP pulse will pull down the graphene ribbon at the central part. Once the graphene ribbon starts to bend, the intrinsic elastic force appears in the graphene membrane. As the distance between the center of the graphene ribbon and the bottom of the cavity decreases, the TLP-induced electrical field force increases to pull down further, while the elastic recovery force in the graphene will also increase. Since the ESD-induced electrostatic force is much stronger than the elastic force, the suspended graphene ribbon will continue to bend until it touches the bottom electrode to turn the gNEMS switch ON for ESD protection. This simulation helps to optimize the gNEMS design for which the ESD-induced pull-down force and elastic recovery force are considered to ensure the gNEMS ESD switching function. After the TLP pulse is over, the ESD-induced electrostatic force will immediately disappear, and the intrinsic elastic force will dominate and pull up the bent graphene ribbon back to its original position, thus turning the gNEMS OFF [16].

FEM simulation can also be used to investigate the stress effect of the suspended graphene ribbon in a gNEMS under ESD zapping and hence provide design guidelines for improving the mechanical reliability of gNEMS devices. As shown in Figure 4, the graphene ribbon was held by metal pads on both ends. During ESD actions, the ESD-induced electrostatic pull-down force creates stress on the graphene ribbon, particularly under the metal pads. In extreme cases, a physical fracture may occur, causing the mechanical failure of a gNEMS structure. To analyze the graphene fracture stress behaviors, various “nails” are designed to “hold” the graphene ribbon at the pad locations, as depicted in Figure 7 [17]. For a comparison study, four nail design splits were designed: a single square nail, a single triangular nail, and four square and triangular nails, which pin down the graphene ribbons with the pads. The idea for the nail design splits was that the nail shape affects the stress, and having more smaller nails may mitigate the stress effects. Figure 8 depicts the fracture stress maps generated through FEM simulation, with the mechanical stress intensity color-coded as blue to indicate the lowest and red to indicate the highest stress pressure. It is readily observed that a single nail induces much heavier stress over the four-nail cases. The single square nail tends to have more stress than its triangular nail counterpart. On the other hand, the case with the four triangular nails is subjected to the lowest stress. Table 1 summarizes the fracture stress results for the four nail designs.

### 2.4. Experiment Results for Poly-Crystalline gNEMS

The gNEMS prototypes were initially designed and fabricated using poly-crystalline graphene ribbons including varying design dimensions [11,18]. Figure 9 depicts the DC-measured I–V characteristics for sample gNEMS devices with varying graphene ribbon lengths of L = 7 μm, 10 μm, 15 μm, and 20 μm, respectively. A diode-like I–V curve shows the turn-on feature of the gNEMS devices. It is also observed that the turn-on voltage is dependent upon the graphene ribbon length, which is reasonable since a longer graphene ribbon undergoes a stronger pull-down force and a weaker elastic force and hence has more potential to bend, leading to a lower turn-on voltage. The current compliance was set to 0.1 mA in DC testing to avoid device failure.

A transient ESD stress test was then conducted using a TLP tester featuring a rise time of 10 ns and a pulse width of 100 ns. Figure 10 depicts the measured ESD I–V characteristics for a sample gNEMS device (d = 350 nm, L = 7 μm, W = 5 μm) under both TLP stressing directions. The transient ESD behavior is clearly achieved. More importantly, the I–V characteristics of dual-directional ESD are observed for the gNEMS devices, which is a unique feature of this gNEMS ESD switch. The slight difference in the I–V curves in two opposite directions is attributed to the imperfection of the gNEMS prototype, as shown in Figure 4. TLP testing confirms that the gNEMS stays OFF until the TLP pulse increases to a certain high level, which will quickly trigger the gNEMS into a low-R discharging mode for ESD protection. Experimental results show that the ESD-triggering voltage (V_t1_) can be adjusted by device design parameters, including cavity depth, as well as the width, length, and shapes of graphene ribbons. The measured leakage current is very low, ~3–13 pA. This gNEMS can handle very high ESD current of up to ~10^8^ A/cm^2^, equivalent to ~1.5 KV/μm^2^, which is much higher than ~7.5 V/μm^2^ for an SCR ESD device (normally considered the most robust ESD protection devices in traditional PN-based structures). It is worth noting that gNEMS supports ESD in both directions, which can dramatically reduce the total ESD head count on a chip and thus significantly reduce the problem of ESD protection design overhead.

The temperature dependence of the ESD behaviors of gNEMS was investigated via TLP testing. Figure 11 shows ESD I–V curves for a sample gNEMS device (L = 10 μm and W = 3 μm) under TLP stress at different temperatures, i.e., T = −10 °C, 30 °C, and 110 °C. It is readily observed that the gNEMS behavior is sensitive to temperature. As temperature increases, V_t1_ decreases. Higher temperature also affects the current handling capability of gNEMS due to thermally induced defects in the graphene membrane [16].

### 2.5. Experiment Results for Single-Crystalline gNEMS

Since the material properties of graphene films will affect gNEMS device performance, single-crystalline graphene was developed to improve the performance of gNEMS devices. A comparison study was carried out for poly-crystalline and single-crystalline graphene gNEMS devices. Figure 12 depicts the I–V characteristics of ESD for the single-crystalline gNEMS devices under both TLP and very-fast TLP (VFTLP) stress tests. Figure 12a presents the DC sweeping test result for a sample gNEMS (L = 5 μm and W = 3 μm) where a diode-like turn-on I–V curve is clearly observed with the turn-on voltage of ~2.45 V. Figure 12b shows the I–V curve of ESD under TLP testing (t_r_ = 10 ns and t_d_ = 100 ns) for the evaluation of human body model (HBM) ESD. The transient ESD I–V curve is observed with V_t1_~7.79 V and I_t2_~30.3 mA. The leakage of I_leak_ ~2 pA is negligible. Figure 12d depicts the ESD voltage and current behaviors in the time domain during TLP stress, based on which the ESD response time (t_1_) of gNEMS can be obtained. Figure 12c illustrates the I–V curve of ESD for gNEMS under ultra-fast VFTLP stress test (t_r_ = 100 ps and t_d_ = 1 ns), which clearly shows that the gNEMS can respond to the ESD pulses of the ultra-fast-charged device model (CDM). According to the VFTLP test results, V_t1_~4.2 V and I_t2_~31.3 mA are observed for the gNEMS device [12,16].

A comparison of the ESD characteristics of single-crystalline and poly-crystalline gNEMS devices is given in Figure 13. In both DC sweeping and TLP stress tests, it is revealed that single-crystalline gNEMS outperforms its poly-crystalline counterpart, for example, with I_t2_~0.37 mA for single-crystalline gNEMS over I_t2_~0.14 mA for poly-crystalline gNEMS in the DC test. This is also reflected in I_t2_~31.1 mA for single-crystalline gNEMS over I_t2_~5.88 mA for poly-crystalline gNEMS in the TLP stress test [12]. The performance enhancement of single-crystalline gNEMS is mainly attributed to the better material properties of single-crystalline graphene films as a result of which defects are dramatically reduced, and the crystalline grain improves both the electrical and thermal conductivity of graphene.

The reduced defect density in single-crystalline graphene can significantly improve the reliability of gNEMS devices, which was confirmed through repeated ESD stress tests, and their results are shown in Figure 14. Figure 14a depicts the results of the DC sweeping test repeated 11 times for a single-crystalline gNEMS where the DC sweeping voltage is clamped below the thermal breakdown current threshold (~0.24 mA) to avoid device failure. This repeated testing approach ensures that the same gNEMS sample can be used for repeating tests, thus increasing the reliability of the analysis results. It is readily observed that the I–V curves of the DC turn-on remain unchanged during the 11 times that the DC sweeping tests were repeated, indicating the good device reliability of the gNEMS switch. Figure 14b shows that the single-crystalline gNEMS sample device has very stable ESD I–V characteristics after repeating the TLP stress test 110 times, which again confirms that the single-crystalline gNEMS device is very stable due to excellent crystalline graphene properties. During the repeated TLP stress tests, the TLP pulse was limited to under the thermal breakdown current (10 mA) to avoid device failure. For clarity, the I–V curves in Figure 14b only show those after every 10 repeated stresses. Similarly, Figure 14c shows ESD I–V characteristics for a sample single-crystalline gNEMS device subjected to the VFTLP stress test 110 times, which again clearly confirms that the gNEMS device is very stable owing to the good properties of single-crystalline graphene films [12].

The impacts of the dimension of graphene ribbons on gNEMS performance were investigated using gNEMS devices designed with varying widths and lengths of graphene membranes. Both TLP and VFTLP measurements were conducted for a large number of gNEMS samples for statistical analysis. Figure 15a depicts the statistical results of V_t1_ ~ W for gNEMS devices of fixed L = 10 μm and varying W (3 μm, 5 μm, 10 μm, and 15 μm) under TLP stress. It is readily observed that V_t1_ is not affected by the width variation because of the counter effect of an increase in both the electrostatic pulling force and the intrinsic elastic force as W increases. Figure 15b shows V_t1_ ~ L of different gNEMS devices with a fixed W = μm and varying L (5 μm, 7 μm, 10 μm, 15 μm, and 20 μm). The TLP test results clearly show that V_t1_ monotonously decreases as L increases, because a longer graphene ribbon undergoes a stronger electrostatic pulling force while experiencing a weaker intrinsic elastic force, making it easier for the suspended graphene membrane to touch the bottom. Similarly, Figure 15c,d shows that, under VFTLP stress, V_t1_ is not affected by W but decreases for a longer L.

The impacts of graphene ribbon dimensions on the ESD current handling capability were also studied using a large number of gNEMS devices for statistical analysis, as depicted in Figure 16. Figure 16a shows the TLP-measured I_t2_~W~L statistics for sample gNEMS, while Figure 16b illustrates the same statistics for the VFTLP test. The test results show that I_t2_ data range from 25.5 mA to 69 mA under TLP stress and from 27.6 mA to 59.9 mA for VFTLP results. It is readily observed that as W increases, at a fixed L, I_t2_ substantially increases, implying that a wider graphene ribbon can handle more ESD current without overheating due to the reduced resistance. On the other hand, at a fixed W, I_t2_ is generally not affected by L, and the slight decrease in I_t2_ for longer L may be because a longer graphene ribbon may have more defects due to the imperfections generated during graphene growth. Figure 16c,d show the highest I_t2_ record measured in TLP and VFTLP tests for single-crystalline sample gNEMS devices (W/L = 7µm/20µm), i.e., I_t2_ ~293 mA under TLP stress, or, the ESD current handling capability of J_t2_~1.19 × 10^10^ A/cm^2^. Briefly, this is equivalent to an HBM ESD capability of ~178 KV/µm^2^, which is much improved compared with that of poly-crystalline gNEMS, at J_t2_~1.5 KV/µm^2^; this is attributed to improved graphene properties in the single-crystalline structure. It is noteworthy that gNEMS is much more ESD-robust than any in-Si PN-based conventional ESD protection structures, e.g., J_t2_~7.5 V/µm^2^ for a typical SCR ESD protection device. A record high I_t2_ of ~149 mA, i.e., J_t2_~6.09 × 10^9^ A/cm^2^, was also obtained under VFTLP stress, indicating the superior CDM ESD protection capacity of gNEMS devices [12].

## 3. Graphene ESD Interconnects

### The Advantages of Graphene ESD Interconnects

It is well known that metal interconnects are weak points in ESD protection. Using wider metal interconnects for ESD protection can improve ESD robustness; however, this will also induce significant parasitic C_ESD_ associated with the metal wires. It seems that the excellent material properties of graphene, such as superior mobility, high thermal conductivity, and mechanical strength, can be very beneficial for ESD protection in IC interconnects. For example, the maximum current handling capability of graphene ribbon (GR) is I_max_~10^8^ A/cm^2^, ten times higher than that of copper, which is currently widely used for IC interconnects. The graphene thermal conductivity of κ = 4.84~5.3 × 10^3^ W/m∙K is about thirteen times higher than copper, greatly mitigating the problem of ESD-induced overheating. Therefore, graphene ribbons were proposed to replace Cu interconnects for local ESD protection circuits [19]. More benefits are expected from using graphene ESD interconnects. Due to high mobility, a graphene ribbon of the same width can handle higher ESD currents, meaning higher levels of ESD protection. On the other hand, for a targeted ESD protection level, less graphene ribbon is needed, compared with a Cu wire, which leads to reduced ESD-wire-induced C_ESD_. Figure 17a shows the concept of using graphene ribbons for ESD protection interconnects. A large number of sample graphene ribbon wires were fabricated for characterization. Figure 17b shows an SEM image of a sample GR wire. The GR samples were fabricated using a new CMOS-compatible process, with the key steps shown in Figure 17c. LPCVD was used to deposit a SiO_2_ layer 250 nm in thickness on a wafer, used as the dielectric layer, to isolate the graphene interconnects. Next, graphene films developed using the CVD method were transferred on top of the wafer, followed by oxygen patterning to create the GR wires. Lastly, an e-beam was used to form two metal pads for measurements.

Comprehensive TLP and VFTLP measurements were carried out for a large number of GR samples. Figure 18a depicts the measured typical ESD I–V curves for a graphene ribbon sample (*L* = 12 µm, *W* = 5 µm) in TLP and VFTLP tests, respectively, showing a critical current density (i.e., the maximum current handling capability) of *J*_C_~10^8^ A/cm_2_, indicating the ultra-high ESD robustness of graphene ribbon wires. Figure 18b statistically compares the measured maximum sustainable power (*P*_C_ = *I*_C_ × V_C_) via TLP and VFTLP testing for graphene ribbon wires of *L* = 7 µm ~ 50 µm at fixed W = 5µm. The results reveal that *P*_C_ increases as *L* increases due to an increase in graphene ribbon resistance with a longer length. Figure 18c reveals that the maximum current (*I*_C_) for bilayer GR wire samples is very sensitive to temperature, and an optimal temperature seems to exist, suggesting an optimum treating condition for GR wires (i.e., *T* ≈ 50 °C–60 °C). Figure 18d shows a similar *I*_C_–T relationship for monolayer GR wire samples.

The impacts of the dimensions of graphene ribbons on the ESD handling capability were studied, with the measured statistics shown in Figure 19. Figure 19a shows the statistics for the critical (ESD failure) voltage *V_C_* and the critical current density *J_C_* related to the length of graphene ribbon samples (at fixed width) in the TLP test. It is readily observed that V_C_ monotonically increases as L increases because of the increased resistance. J_C_ seems to be unaffected by L because *I*_C_ is normalized to W. Similar trends are observed for GR samples subjected to stress in the VFTLP test, as depicted in Figure 19b. Figure 19c shows the statistics for *I*_C_ and *J*_C_ related to the width of graphene ribbon samples (at fixed length) in the TLP test. It is clear that *I*_C_ increases as W becomes wider because of the reduced resistance, while *J*_C_ is unaffected by W. Similar trends are observed for GR samples subjected to stress in the VFTLP test, as depicted in Figure 19d.

The ESD failure mechanism of GR wires was studied using time-resolved Raman spectroscopy by continuously subjecting GR samples to stress using the TLP method until ESD failure would occur. Figure 20a shows the Raman scanning image for a GR sample before and after TLP zapping; the Raman D-peak intensity is an indicator of defect accumulation within the stressed GR wire. The process of ESD failure development is readily observed in Figure 20a, showing that, before TLP stress, defects are mainly located along the boundaries of CVD-generated graphene, while after being subjected to TLP stress, defect localization occurs, leading to the formation of a fault line across the graphene ribbon wire (the white dashed line). It has been recently suggested that the study of the intensity ratio of G and D peaks can reveal more details on graphene reliability [20]. Figure 20b shows the failure signature of a graphene ribbon wire subjected to stress via TLP, in which the presence of a crack (red dashed circle) indicates the ESD failure signature.

## 4. Summary

On-chip ESD protection is required for ICs. As IC technology continuously advances, and IC complexity rapidly increases, the design of ESD protection becomes more and more challenging because the ESD-induced design overhead effect becomes a major design barrier. Conventional ESD protection solutions rely on in-Si PN-based ESD protection devices and their derivatives, which unfortunately induce significant ESD parasitic effects, which are becoming unacceptable to modern ICs in advanced technology nodes. To fundamentally overcome the existing ESD protection design challenges, a novel concept comprising a disruptive graphene-based gNEMS ESD switch and graphene ribbon ESD interconnects was proposed for future ESD protection designs. This paper presented the gNEMS device concept and experimental results. The gNEMS prototypes validated the novel gNEMS ESD switch structure. It is noteworthy that, while the obtained results confirm the reliability of the new gNEMS ESD protection device, more research is needed to optimize the new device structure, including fabrication processes and characterization methods, e.g., leveraging recent advances in fabricating locally suspended few-layer graphene nanostructures via irradiation [20] and transfer-free graphene development techniques [21]. This review aims to inspire non-traditional thinking for on-chip ESD protection designs, possibly leading to revolutionary future ESD protection solutions.

## Figures and Tables

**Figure 1 nanomaterials-13-01426-f001:**
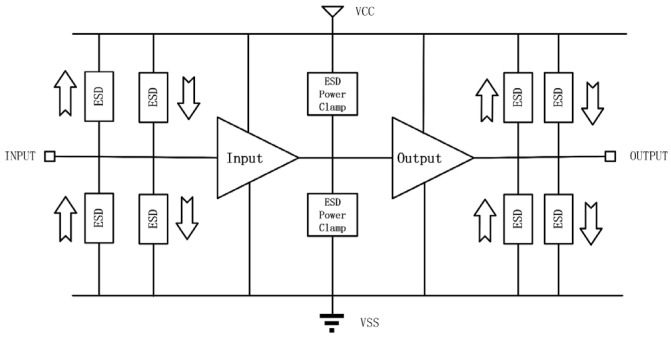
Illustration for a full-chip ESD protection scheme.

**Figure 2 nanomaterials-13-01426-f002:**
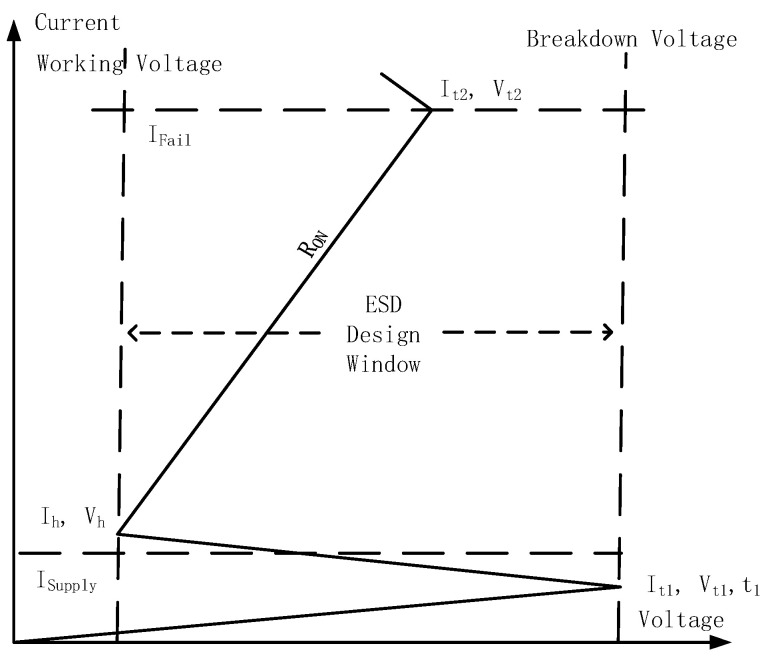
Typical ESD I–V characteristics and ESD design window.

**Figure 3 nanomaterials-13-01426-f003:**
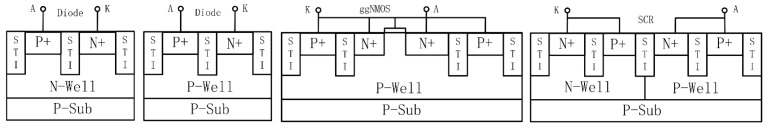
Traditional in-Si PN-based ESD protection structures (A = anode and K = cathode).

**Figure 4 nanomaterials-13-01426-f004:**
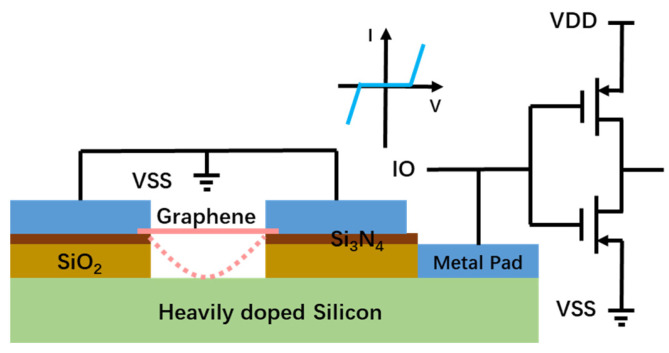
Cross-section of gNEMS ESD protection device in IC connection. Inset shows the symmetric ESD I–V characteristics.

**Figure 5 nanomaterials-13-01426-f005:**
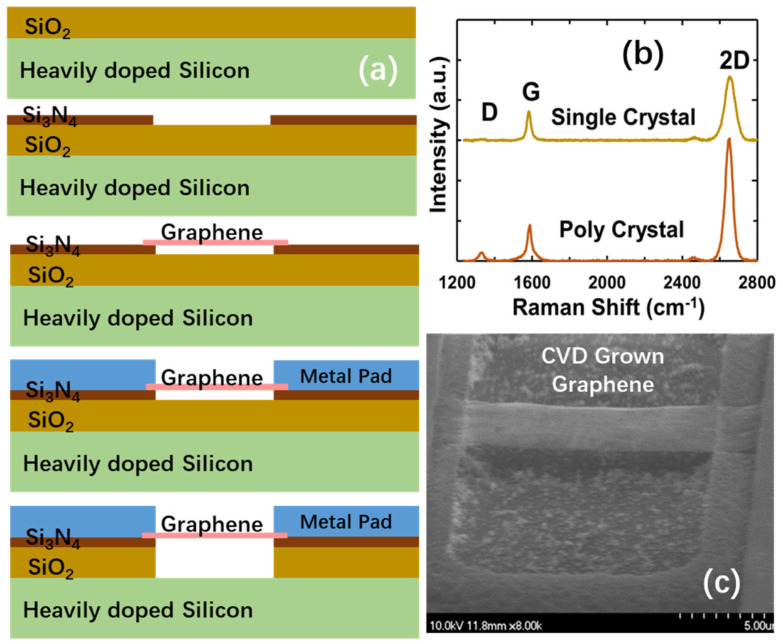
gNEMS fabrication process: (**a**) CMOS-compatible fabrication procedure; (**b**) Ramen spectrum confirming graphene materials and distinguishing single-crystalline and poly-crystalline graphene structures; (**c**) SEM image of gNEMS structure showing the suspended graphene ribbon over the cavity.

**Figure 6 nanomaterials-13-01426-f006:**
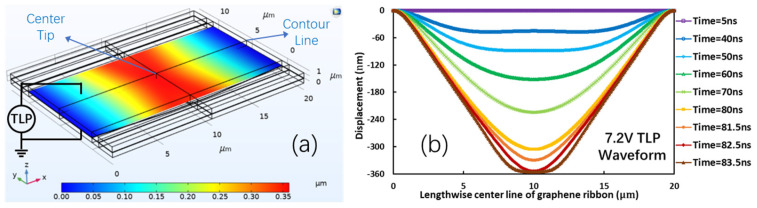
(**a**) The 3D transient FEM simulation shows vertical displacement of the graphene ribbon in a gNEMS of L = 20µm, W = 10µm, and d = 350 nm at the moment of contact. Physical displacement in *Z*-axis is color-coded with blue for zero displacement at z = 0 and red for the largest bending of z = −350 nm at the moment of contact; (**b**) simulated graphene ribbon’s vertical displacement resulting from the stress induced by a 7.2 V TLP pulse for a sample gNEMS reveals the t-dependence of the triggering behaviors of gNEMS.

**Figure 7 nanomaterials-13-01426-f007:**
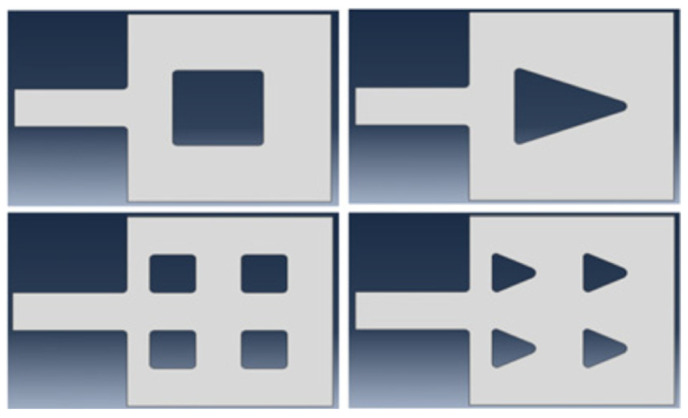
Four gNEMS nail design splits, including one square nail, one triangular nail, four smaller square nails, and four smaller triangular nails, for a comparison study of fracture stress in graphene ribbons in gNEMS devices.

**Figure 8 nanomaterials-13-01426-f008:**
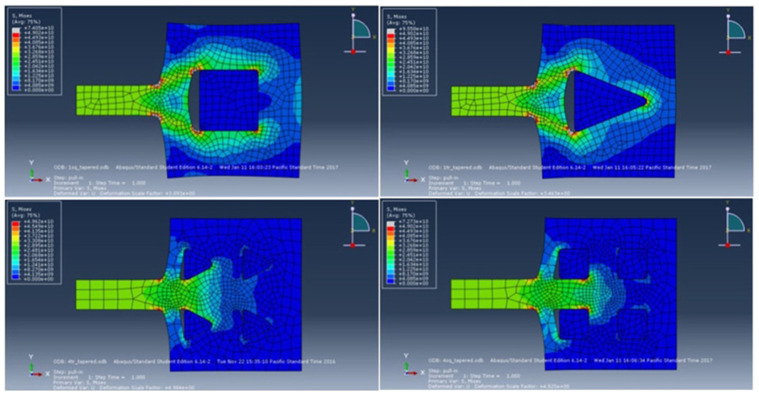
FEM simulations of fracture stress in graphene ribbons in gNEMS devices with four nail designs: single square nail, single triangular nail, four smaller square nails, and four triangular nails.

**Figure 9 nanomaterials-13-01426-f009:**
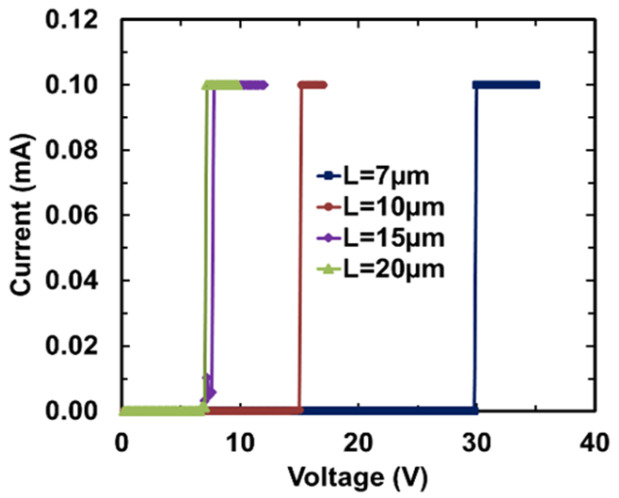
Samples of gNEMS devices with L = 7 μm, 10 μm, 15 μm, and 20 μm show the turn-on behavior at different DC sweeping voltages.

**Figure 10 nanomaterials-13-01426-f010:**
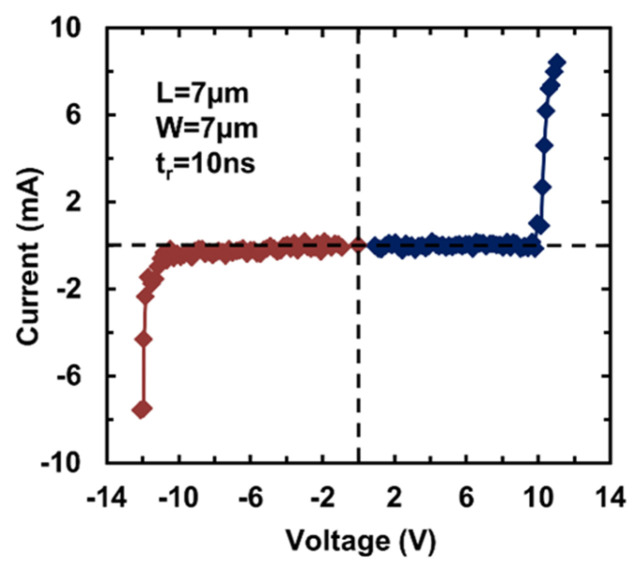
A sample gNEMS switch shows the I–V characteristics of dual-directional ESD under TLP testing.

**Figure 11 nanomaterials-13-01426-f011:**
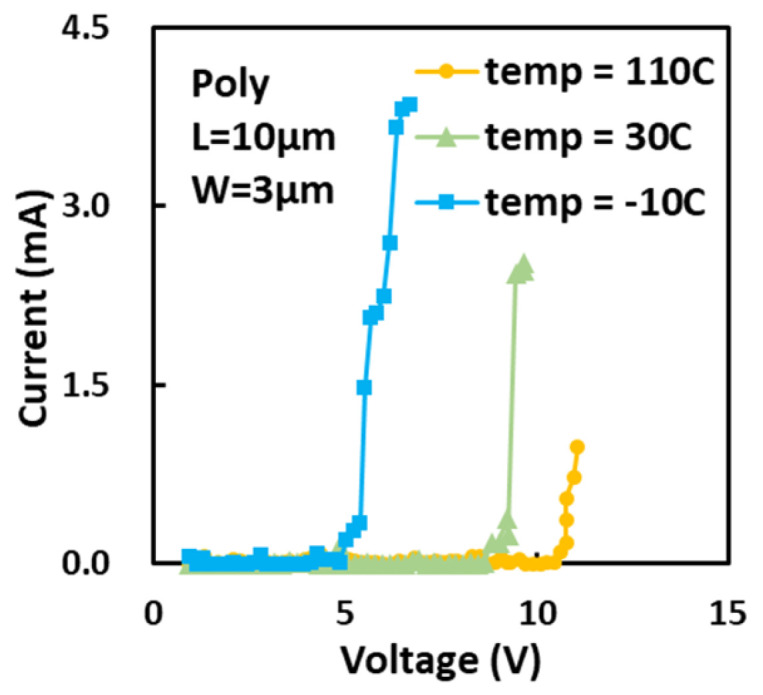
Temperature effects of a sample poly-crystalline gNEMS ESD switch via TLP testing.

**Figure 12 nanomaterials-13-01426-f012:**
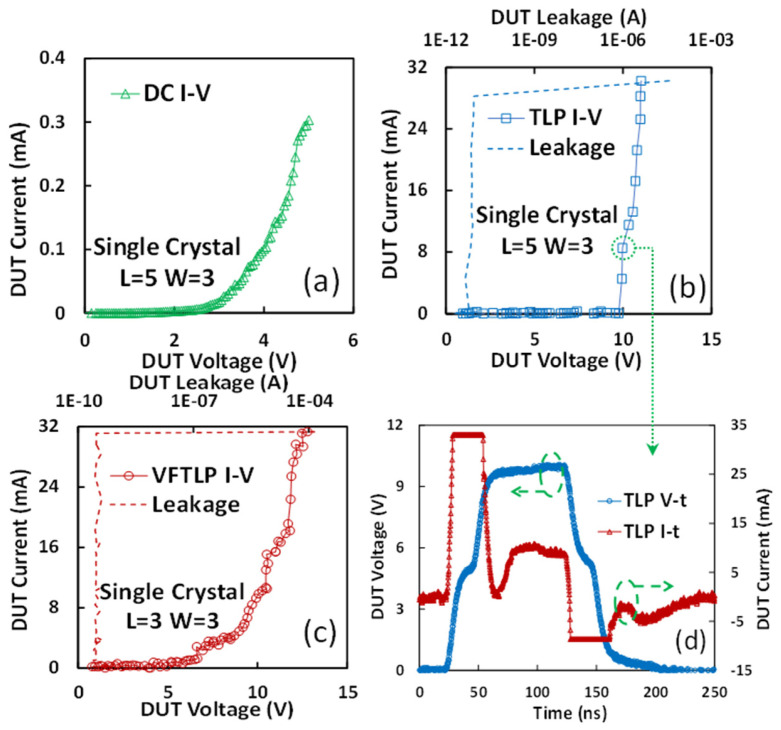
Measured switching I–V behaviors for different gNEMS devices using different test methods: (**a**) DC sweeping; (**b**) TLP stress for the HBM ESD model; (**c**) VFTLP stress for the CDM ESD model; (**d**) ESD I–V waveforms in the t-domain under TLP testing.

**Figure 13 nanomaterials-13-01426-f013:**
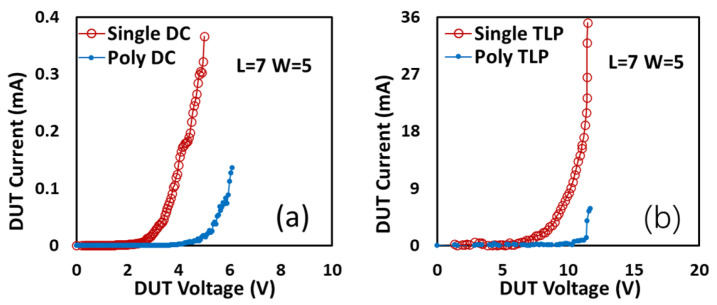
Comparison between single-crystalline and poly-crystalline graphene gNEMS switches: (**a**) DC sweeping I–V curves; (**b**) transient ESD I–V curves under TLP zapping.

**Figure 14 nanomaterials-13-01426-f014:**
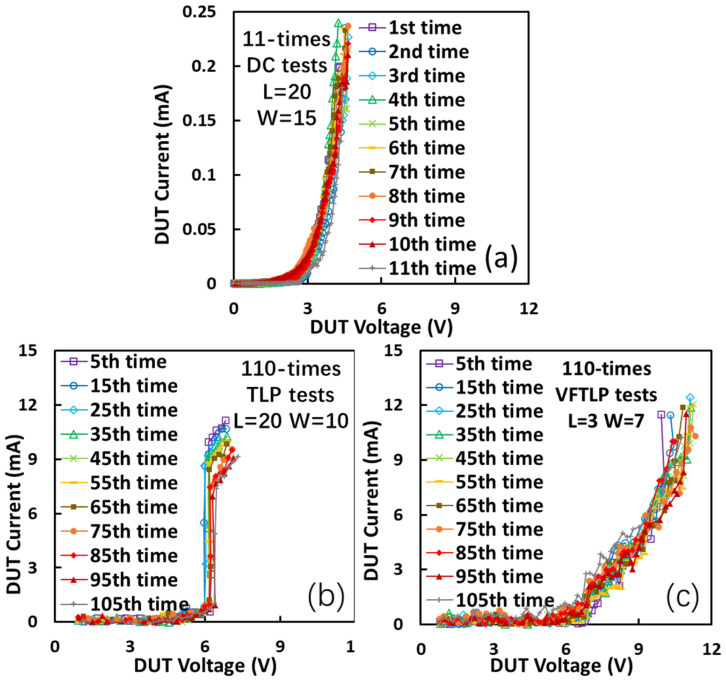
Repeated ESD stress tests for different single-crystalline gNEMS devices of varying sizes confirm that the gNEMS structures are extremely stable: (**a**) DC sweeping stress test repeated 11 times; (**b**) TLP zapping test repeated 110 times; (**c**) VFTLP zapping test repeated 110 times.

**Figure 15 nanomaterials-13-01426-f015:**
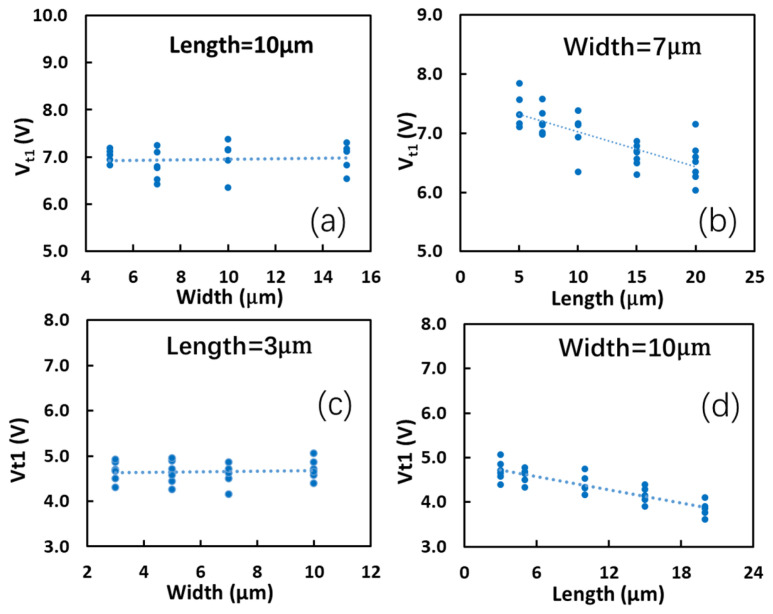
Statistical results for impacts of graphene ribbon dimensions on ESD V_t1_ in tests: (**a**) V_t1_~W at a fixed L is almost flat under TLP; (**b**) V_t1_~L at a given W shows a monotonous trend under TLP; (**c**) V_t1_~W at a fixed L is almost flat under VFTLP; and (**d**) V_t1_~L for a given W shows a monotonous trend under VFTLP.

**Figure 16 nanomaterials-13-01426-f016:**
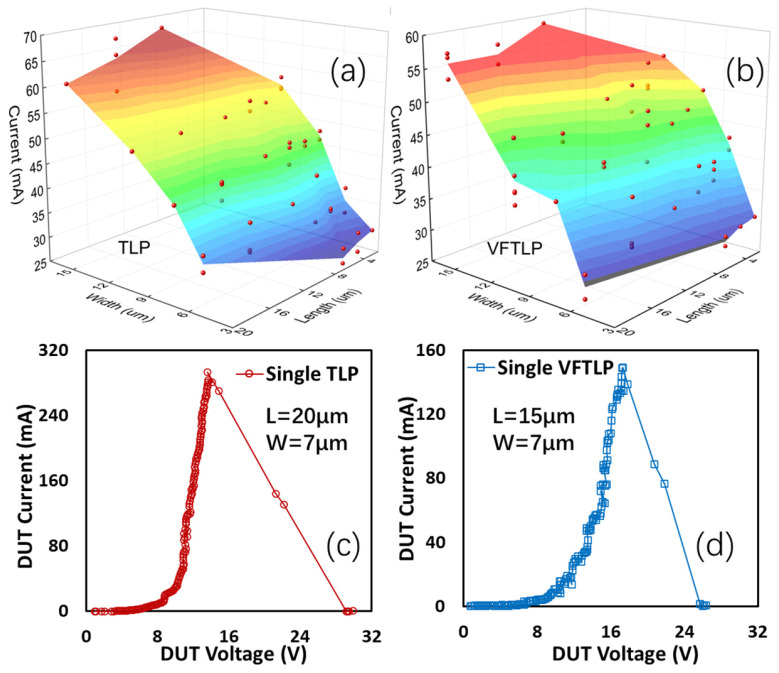
ESD I_t2_ capability of different gNEMS devices of varying dimensions under TLP and VFTLP stresses: (**a**) results of TLP zapping, (**b**) results of VFTLP stress test; the record-setting I_t2_ for single-crystalline gNEMS under (**c**) TLP stress and (**d**) VFTLP stress.

**Figure 17 nanomaterials-13-01426-f017:**
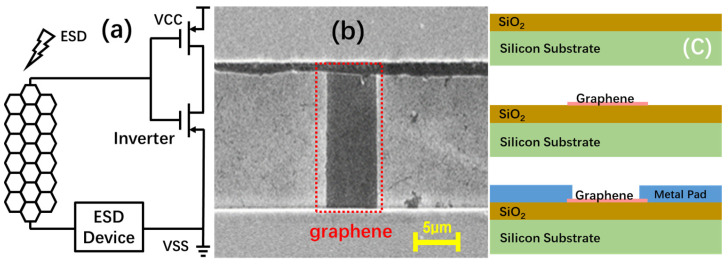
Graphene ESD interconnects: (**a**) an on-chip ESD protection scheme using GR wires as alternatives to metal wires; (**b**) SEM image for a GR sample with L = 12 µm and W = 5 µm; (**c**) CMOS-compatible fabrication in 3 main steps.

**Figure 18 nanomaterials-13-01426-f018:**
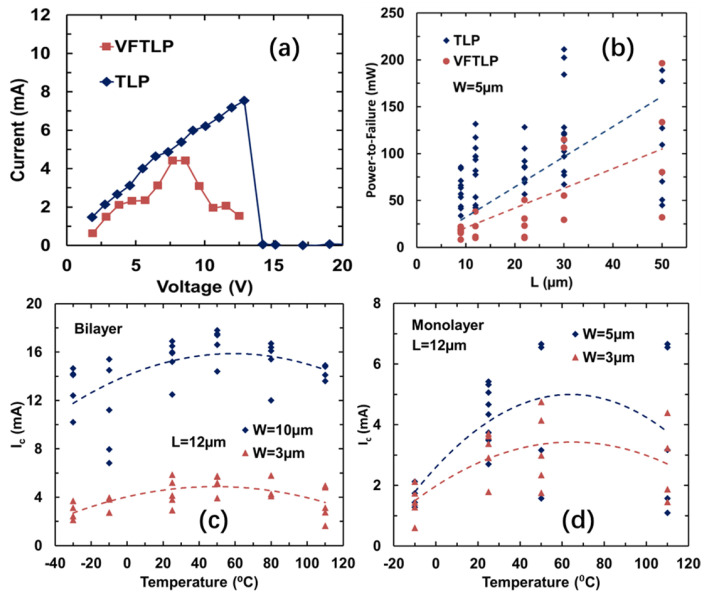
Measurement of graphene ESD interconnects: (**a**) measured ESD I–V curves for a bilayer GR sample (*L* = 12 µm, *W* = 5 µm) in TLP and VFTLP tests; (**b**) statistics of P_C_–L relationship of GR samples under TLP and VFTLP stresses; (**c**) bilayer graphene interconnects under TLP stress with varying temperature from −30 °C to 110 °C; (**d**) monolayer graphene interconnects under TLP stress with varying temperature from −30 °C to 110 °C.

**Figure 19 nanomaterials-13-01426-f019:**
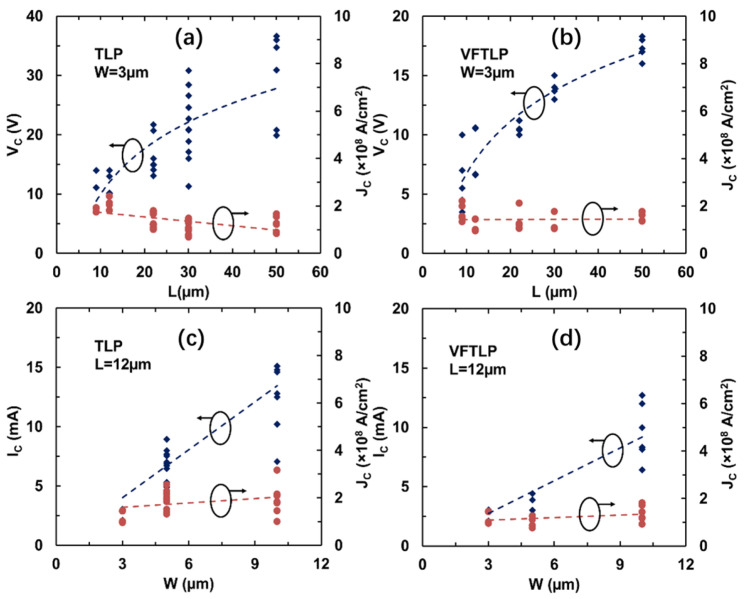
Statistics for measured *V*_C_ and *J*_C_ versus *L* and *W* for GR wires: (**a**) TLP for *W* = 3 µm; (**b**) VFTLP for *W* = 3 µm; (**c**) TLP for *L* = 12 µm; (**d**) VFTLP for *L* = 12 µm.

**Figure 20 nanomaterials-13-01426-f020:**
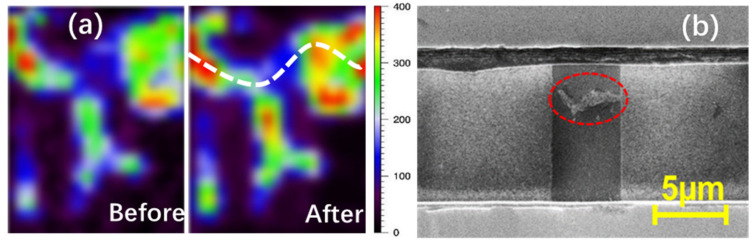
(**a**) D-peak intensity in a Raman spectrum for a graphene ribbon sample before and after TLP zapping failure, which illustrates defect development, leading to ESD failure; (**b**) SEM image shows the failure signature, which is a fault line across the graphene wire after ESD zapping breakdown.

**Table 1 nanomaterials-13-01426-t001:** Summary of the simulated fracture stress for the four nailed gNEMS devices.

Nail Structure	Maximum Stress (GPa)	Maximum Stress to Pull-in Stress Ratio
One square nail	74.04	2.468
One triangular nail	95.58	3.186
Four square nails	72.73	2.424
Four triangular nails	49.02	1.643

## Data Availability

This is a review paper not offering any original data.

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
