# Peer review of "Graphene-Based ESD Protection for Future ICs"

_nanomaterials, 2023, doi:10.3390/nano13081426_

Round 1
Reviewer 1 Report
This paper shows new type of ESD protection using GNEMS. On-chip ESD protection for ICs, traditional Si devices have disadvantage like leakage current, parasitic capacitance and etc. The possibility of using GNEMS to solve these problems was demonstrated.
1. In the figure 5, graphene is already suspended after transfer. Is it real? When the graphene transfer process (I think this process maybe wet transfer), graphene is fully covered following the substrate surface. So, graphene is contact with SiO2. That means air gap is 250nm, not 350nm.
2. In the figure 9, could you show the log scale curve? The advantage of NEMS device is very low leakage current and abrupt switching process. In the linear curve, subthreshold swing looks high compared theoretical value.
3. Using the vapor HF process to etching the SiO2, are there any problems with graphene damage or contact between graphene and SiO2 due to capillary action?
4. What is the level of damage to graphene when it falls after contact with Si during the repeated operation of GNEMS? Also, please clarify whether the damage of graphene is contact-related or if there are other causes.
5. Because of difference of electrostatic force and elastic force in graphene, pull-in voltage and pull-down voltage are different. In this device, could you show the pull-in and pull-down operation?
Author Response
Dear Reviewer:
Thank you for your review comments. We provide a response as attached.
Regards
Authors

Reviewer 2 Report
The paper is a self-review work. 19 papers are in the references list and 14 are the works of the authors themselves - this seems to be too much. Only one ESD protection method is described and analyzed in detail. A review work should have a broader view on the subject. A list of abbreviations is required, as literally every sentence contains several abbreviations, which makes reading difficult sometimes. Otherwise, the topic is up-to-date, the presentation is quite clear and comprehensible, the paper may be of interest to many readers of the journal.
Author Response

(The authors gave the same response as above.)

Reviewer 3 Report
Referee report
The article “Graphene-Based ESD Protection for Future ICs” is devoted to quite actual problem - electrostatic discharge protection. It is interesting topic. The authors proposed an original approach based on application of graphene ribbons. Metal based NEMS devices are known to face problems, the main one being "sticking" of contacts when high current flows. It is possible that graphene-based devices will not have this disadvantage. It is known that graphene can be stretched and deformed, it is possible to manufacture “suspended” membranes based on graphene, see for example - N.A. Nebogatikova, et. al. “Visualization of Swift Ion Tracks in Suspended Local Diamondized Few-Layer Graphene” Materials, 16, 1391 (2023). DOI: https://doi.org/10.3390/ma16041391. The article contains new experimental data and will be interesting for researchers and technologists. The article can be published, the author can take into account my comments at will.
Comment:
1) The authors have mapped the intensity of the D-peak from graphene (Figure 20), however, to determine the defectiveness of graphene, it is important not the absolute intensity of this peak itself, but the ratio of the intensities of the D and G peaks (for example, as in the already cited work by Nebogatikova et al.). In addition, it was possible to try to determine the mechanical deformations in the graphene membrane analyzing the shift of the G-peak. But this is more of a wish to the authors for the future.
2) All abbreviations should be explained, for example - BJTs, MOSFETs.
Author Response

(The authors gave the same response as above.)

Round 2
Reviewer 2 Report
Authors, thank you for your response. No more comments.